# Cost-utility analysis of proton beam therapy for locally advanced esophageal cancer in Japan

Takuya Sawada[1], Masahide Kondo[2]\*, Masaaki Goto[1,3], Motohiro Murakami[1], Toshiki Ishida[1], Yuichi Hiroshima[1,4], Shu-Ling Hoshi[2], Reiko Okubo[2,5], Toshiyuki Okumura[1,4], Hideyuki Sakurai[1]

1 Department of Radiation Oncology & Proton Medical Research Center, Faculty of Medicine, University of Tsukuba, Tsukuba, Ibaraki, Japan, 2 Department of Health Care Policy and Health Economics, Faculty of Medicine, University of Tsukuba, Tsukuba, Ibaraki, Japan, 3 Department of Radiation Oncology, Japanese Red Cross Medical Center, Tokyo, Japan, 4 Department of Radiation Oncology, Ibaraki Prefectural Central Hospital, Kasama, Ibaraki, Japan, 5 Department of Clinical Laboratory Medicine, University of Tsukuba Hospital, Tsukuba, Ibaraki, Japan

\* mkondo@md.tsukuba.ac.jp

**Data Availability Statement:** All data generated or analysed during this study are included in this published article.

## Abstract

### Purpose

Proton beam therapy (PBT) has recently been included in Japan's health insurance benefit package for certain cancer types. This study aimed to determine the cost-effectiveness of PBT as a replacement for conventional three-dimensional conformal radiotherapy (3D-CRT) for locally advanced esophageal cancer (LAEC) that is not covered by social insurance.

### Methods

We estimated the incremental cost-effectiveness ratio (ICER) of PBT as a replacement for 3D-CRT, using clinical evidence from the literature and expert opinions. We used an economic model, decision tree, and Markov model to illustrate the courses followed by patients with LAEC. Effectiveness was estimated as quality-adjusted life years (QALY) using utility weights for the health state. Social insurance fees were calculated as costs. We assumed two base cases depending on the two existing levels of fees for PBT in social insurance: 2,735,000 Japanese yen (US$20,652) or 1,600,000 yen (US$13,913). The stability of the ICER against these assumptions was appraised using sensitivity analysis.

### Results

The effectiveness of PBT and 3D-CRT was 2.62 and 2.51 QALY, respectively. The estimated ICER was 14,025,268 yen (US$121,958) per QALY for the higher fee level and 7,026,402 yen (US$61,099) for the lower fee level. According to the Japanese threshold for cost-effectiveness of anticancer therapy of 7,500,000 yen (US$65,217) per QALY gain, the inclusion of PBT for LAEC in the benefit package of social insurance is cost-effective if a lower fee is applied.

**Funding:** The author(s) received no specific funding for this work.

**Competing interests:** The authors have declared that no competing interests exist.

## Conclusion

PBT is a cost-effective alternative to 3D-CRT for LAEC and making it available to patients under social insurance could be justifiable.

## 1. Introduction

Chemoradiotherapy (CRT) for locally advanced esophageal cancer (LAEC) is the standard treatment when surgery is not feasible or declined by the patients [1, 2]. Since the number of patients who are intolerant to surgery or have many complications is increasing with the aging population, CRT for LAEC will continue to play an important role. However, CRT also has the potential for adverse events such as radiation pneumonitis (RP) and pericardial effusion (PCE), which may reduce the quality of life (QOL) [3].

Two radiotherapy methods are used in CRT: X-ray radiotherapy (XRT) and particle beam therapy. XRT includes three-dimensional conformal radiotherapy (3D-CRT) and intensity-modulated radiotherapy (IMRT), particle beam therapy includes proton beam therapy (PBT) and carbon-ion radiotherapy. In Japan, 3D-CRT is the standard radiotherapy method with strong evidence for definitive CRT for LAEC [3, 4]. However, PBT has been reported to have a slightly better overall survival (OS) with fewer adverse events [5] and is becoming increasingly popular as a highly advanced medical technology and treatment of choice.

Other treatment modalities for LAEC such as IMRT and carbon-ion radiotherapy have not been frequently reported. Although a few cases of IMRT have been reported, these are often limited to small subsets of esophageal cancer, such as cervical esophageal cancer [6, 7], and some exercise caution regarding its implementation in thoracic esophageal cancer owing to the risk of RP and the need for further research [8, 9]. In addition, the Japanese guidelines for the treatment of esophageal cancer recommend PBT over IMRT as the treatment of choice for patients with poor cardiopulmonary function [10]. Therefore, we suggest that PBT be considered an alternative to standard treatments.

However, compared with 3D-CRT, PBT is more expensive because of the cost of the accelerator and beam delivery system [11]. PBT for carcinomas such as prostate cancer was first covered by Japan's social insurance in 2018. Currently, the indication is approved for eight different cancers, and the fee is set at either 1,600,000 yen (US$13,913; US $1 = 115 yen [\]) or 2,375,000 yen (US$20,652) depending on the type of cancer. PBT for LAEC has the advantage of reduced cardio and pulmonary toxicity [5, 12] but is currently not covered by social insurance and requires self-payment. In Japan, cost-effectiveness has been discussed by the Central Social Insurance Medical Council for expensive medical technology and is an important factor in insurance coverage [13]. This study aimed to demonstrate the cost-effectiveness of PBT compared with 3D-CRT for LAEC.

## 2. Methods

We performed a cost-utility analysis (CUA) of the implementation of PBT in CRT for LAEC in Japan, from the payer's perspective. A literature review was conducted using the PubMed database and the Central Journal of Medicine (Japana Centra Revuo Medicina) to determine the available clinical evidence and cost of introducing PBT. We compared PBT with the standard treatment, 3D-CRT. We then built a decision tree and Markov model to estimate the incremental cost-effectiveness ratio (ICER) of PBT as an alternative to 3D-CRT. These models

were used to illustrate the clinical course of patients with LAEC and were designed based on the opinions of radiotherapy specialists in esophageal cancer at the University of Tsukuba and a radiotherapy specialist at another hospital. Our study was modeled based on only expert opinion and published data.

## 2.1 Target patients

The target population was patients with LAEC Stage II/III (Union for International Cancer Control TNM classification) who required CRT because of difficult surgery or refusal to undergo surgery.

## 2.2 Target treatment and advantage of PBT

The only difference between 3D-CRT and PBT is the type of radiation, with no differences in radiation dose or chemotherapy regimen. In Japan, Ono et al. conducted a nationwide retrospective survey, reporting the results of PBT, and found that PBT showed slightly better OS than XRT [5]. Several studies have suggested that PBT is superior to XRT [14, 15], whereas others have suggested that it is equal to this [16–18]. In this study, the analysis was conducted under more stringent conditions with no difference in OS or progression-free survival (PFS) between 3D-CRT and PBT. For clinical data, we used the JCOG (Japan Clinical Oncology Group) 9906 trial, a prospective study in Japan, for 3D-CRT [3] and data for PBT from Ono et al. [5].

The advantage of PBT is that it reduces adverse events. Differences exist between 3D-CRT and PBT in terms of RP, pleural effusion (PE), and PCE. For example, Grade 3 (Common Terminology Criteria for Adverse Events [CTCAE] ver. 4.0) adverse events of RP, PE, and PCE were reported to be 4%, 9%, and 16%, respectively, for 3D-CRT [3] and 0.5%, 0%, and 1%, respectively, for PBT [5]. Thus, in terms of reduced pulmonary and cardiotoxicities, PBT has an advantage over 3D-CRT for reducing adverse events and improving QOL. Although CTCAE ver. 2.0 was used in the clinical data of 3D-CRT adopted in this study, we considered that it can be replaced by CTCAE ver. 4.0 because it is almost the same as CTCAE ver. 4.0 for RP, PE, and PCE.

## 2.3 Model structure

Fig 1 shows the decision tree and Markov model used in this study. We assumed two alternatives in the decision tree:3D-CRT and PBT. The subsequent clinical courses of the patients were explained using a Markov model. In this model, we described health states that explained adverse events and omitted recurrence and metastasis as we assumed a lack of difference in progressive disease between the two treatments.

The Markov model was started with a health state with no late adverse events after CRT. The health states for the eight types of adverse events were RP Grade 1–3, PE Grade 1–3, and PCE Grade 2–3. The CTCAE Ver 4.0 was used to define adverse event grades (no PCE Grade 1). Frequently overlapping adverse events were also included in the three independent health states. RP Grade 1 + PE Grade 1 or PCE Grade 2 and severe PCE and PE may also overlap, and we assumed that these conditions overlap as well. Health states as a consequence of PCE treatment included drainage, pericardiotomy, and resolved PCE.

Every patient with or without adverse events was moved to best supportive care (BSC), followed by death.

Regarding the transitions after health states for adverse events, among the RP health states, Grade 3 patients could move to Grade 2 or 1 and Grade 2 patients could move to Grade 1 after treatment. Grade 2 and 1 patients could stay. Among the PE health states, Grade 3 patients

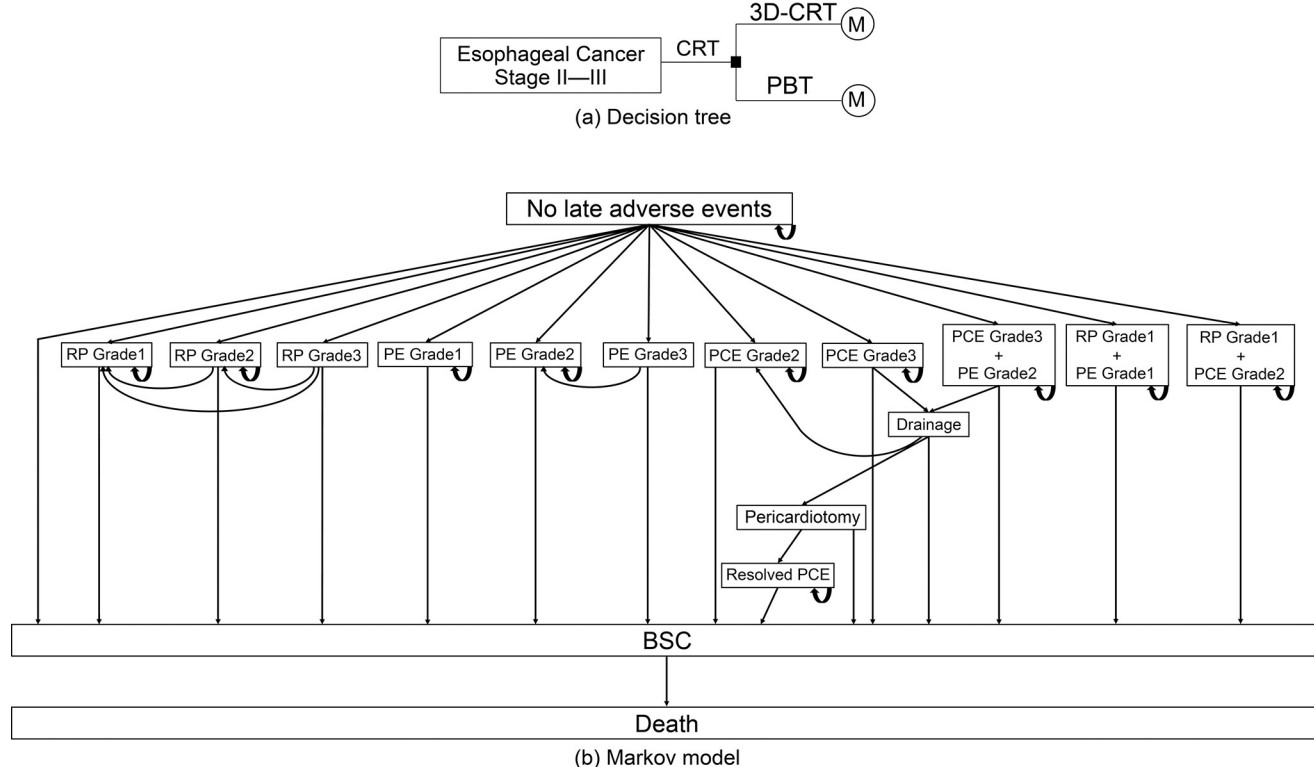

**Fig 1. Models used to estimate the cost-effectiveness of proton beam therapy.** Abbreviations: CRT, chemoradiation therapy; PBT, proton beam therapy; 3D-CRT, three-dimensional conformal radiotherapy; M, Markov model; RP, radiation pneumonitis; PE, pleural effusion; PCE, pericardial effusion; BSC, best supportive care.

improved to Grade 2. The transition to Grade 1 was omitted by expert opinion as diuretics were not withdrawn and continued medication was classified as Grade 2 by definition. Grades 2 and 1 could stay. Among the PCE health states, Grade 3, with and without overlap with PE Grade 2, could move to drainage once heart failure develops. Patients on drainage could move to Grade 2 with successful treatment, whereas unsuccessful patients underwent pericardiotomy, followed by a resolved PCE. Grades 2 and 3 overlapped with PE Grade 2 and resolved PCE could stay. Patients with overlapping states, such as RP Grade 1 with PE Grade 1 or PCE Grade 2 could stay. We assumed that every patient died three months after BSC.

The Markov cycle for each stage was set at three months, with the model programmed to cease when the cohort stage reached the 60th month.

### 2.4 Transition probability

Table 1 presents the transition probabilities. To incorporate the advantages of PBT, the probabilities from no late adverse events after chemoradiation to adverse events were determined based on the JCOG 9906 trial for 3D-CRT [3] and Ono et al. for PBT [5, 19]. The transition probability to RP was assumed to be zero one year after the end of CRT by expert opinion, and that of PE and PCE was set to zero after two years [20, 21]. Although this contradicts theory, using the clinical evidence from Kato and Ono, the incidence of RP Grade 1 is assumed to be higher for PBT than for 3D-CRT [3, 5, 19]. In other studies, the incidence of PR Grade 1 after PBT was similar to that reported by Ono [22]. Theoretically, a major difference between 3D-CRT and PBT is the transition probability to Grade 3 adverse events. For example, for RP

**Table 1. Model assumptions.**

| Variable | Value | PBT | 3D-CRT | Reference |
|---|---|---|---|---|
| Transition probability in Markov Model | | | | |
| Transition probability from any stage to BSC | | 0.0542 | 0.0542 | [3] |
| Transition probability from no late adverse events to PR Grade 1 in 1st year | | 0.095 | 0.046 | [3, 5, 19] |
| Transition probability from no late adverse events to PR Grade 1 after 2nd year | | 0 | 0 | expert opinion |
| Transition probability from no late adverse events to PR Grade 2 in 1st year | | 0.005 | 0.053 | [3, 5] |
| Transition probability from no late adverse events to PR Grade 2 after 2nd year | | 0 | 0 | expert opinion |
| Transition probability from no late adverse events to PR Grade 3 after 1st year | | 0.0025 | 0.026 | [3, 5] |
| Transition probability from no late adverse events to PR Grade 3 after 2nd year | | 0 | 0 | expert opinion |
| Transition probability from RP Grade 3 to RP Grade 2 | | 0.25 | 0.25 | [23] |
| Transition probability from RP Grade 2 to RP Grade 2 | | 0.25 | 0.25 | [23] |
| Transition probability from no late adverse events to PR Grade 1 + PE Grade 1 in 1st year | | 0.157 | 0.179 | [3, 5, 19] |
| Transition probability from no late adverse events to PR Grade 1 + PE Grade 1 after 2nd year | | 0 | 0 | expert opinion |
| Transition probability from no late adverse events to PR Grade 1 + PCE Grade 2 in 1st year | | 0.076 | 0.081 | [3, 5, 19] |
| Transition probability from no late adverse events to PR Grade 1+ PCE Grade 2 after 2nd year | | 0 | 0 | expert opinion |
| Transition probability from no late adverse events to PE Grade 1 by 2nd year | | 0.008 | 0.041 | [3, 5, 19] |
| Transition probability from no late adverse events to PE Grade 1 after 3rd year | | 0 | 0 | [20, 21] |
| Transition probability from no late adverse events to PE Grade 2 by 2nd year | | 0.017 | 0.015 | [3, 5] |
| Transition probability from no late adverse events to PE Grade 2 after 3rd year | | 0 | 0 | [20, 21] |
| Transition probability from no late adverse events to PE Grade 3 by 2nd year | | 0 | 0.06 | [3, 5] |
| Transition probability from no late adverse events to PE Grade 3 after 3rd year | | 0 | 0 | [20, 21] |
| Transition probability from no late adverse events to PCE Grade 2 by 2nd year | | 0.0024 | 0.02 | [3, 5] |
| Transition probability from no late adverse events to PCE Grade 2 after 3rd year | | 0 | 0 | [20] |
| Transition probability from no late adverse events to PCE Grade 3 by 2nd year | | 0.001 | 0.075 | [3, 5] |
| Transition probability from no late adverse events to PCE Grade 3 after 3rd year | | 0 | 0 | [20] |
| Transition probability from PBT to PCE Grade 3 + PE Grade 2 by 2nd year | | 0.003 | 0.028 | [3, 5] |
| Transition probability from PBT to PCE Grade 3 + PE Grade 2 after 3rd year | | 0 | 0 | [20] |
| Transition probability from PCE Grade 3 or PCE Grade 3 + PE Grade 2 to drainage | | 0.1 | 0.1 | [24] |
| Transition probability from drainage to PCE Grade 2 | | 0.617 | 0.617 | [25] |
| Transition probability from BSC to Death | | 1 | 1 | expert opinion |
| Utility Score | Value | | | Reference |
| No late adverse events | | 0.91 | 0.91 | [26] |
| RP Grade 1 | | 0.87 | 0.87 | [26] |
| RP Grade 2 | | 0.83 | 0.83 | [26] |
| RP Grade 3 | | 0.54 | 0.54 | [27] |
| PE Grade 1 | | 0.87 | 0.87 | [26] |
| PE Grade 2 | | 0.76 | 0.76 | [28] |
| PE Grade 3 | | 0.58 | 0.58 | [29] |
| Resolved PCE | | 0.87 | 0.87 | [26] |
| PCE Grade 2 | | 0.87 | 0.87 | [26] |
| PCE Grade 3 | | 0.63 | 0.63 | [30] |
| PCE Grade 3 + PE Grade 2 | | 0.57 | 0.57 | [30] |
| RP Grade 1 + PE Grade 1 | | 0.87 | 0.87 | [26] |
| RP Grade 1 + PCE Grade 2 | | 0.87 | 0.87 | [26] |
| BSC | | 0.32 | 0.32 | [31] |
| Cost | Value | | | Reference |
| PBT (including management fee) | | ¥2,375,000 or ¥1,600,000 | — | Social insurance fee |

(*Continued*)

**Table 1.** (Continued)

| Variable | Value | PBT | 3D-CRT | Reference |
|---|---|---|---|---|
| 3D-CRT (including management fee) | | − | ¥727,600 | Social insurance fee |
| Admission and chemotherapy during chemoradiotherapy (excluding radiotherapy fee) | | ¥1,325,370 | ¥1,325,370 | Social insurance fee |
| Follow-up (examination, blood tests, CT scan, endoscopy) in 1st year | | ¥45,615 | ¥45,615 | Social insurance fee |
| Follow-up (examination, blood tests, CT scan, endoscopy) after 2nd year | | ¥28,480 | ¥28,480 | Social insurance fee |
| Treatment of RP Grade 2 | | ¥62,160 | ¥62,160 | Social insurance fee |
| Treatment of RP Grade 3 | | ¥571,120 | ¥571,120 | Social insurance fee |
| Treatment of PE Grade 2 (including follow-up fee in 1st year) | | ¥47,165 | ¥47,165 | Social insurance fee |
| Treatment of PE Grade 2 (including follow-up fee after 2nd year) | | ¥30,030 | ¥30,030 | Social insurance fee |
| Treatment of PE Grade 3 | | ¥302,700 | ¥302,700 | Social insurance fee |
| Pericardial drainage | | ¥448,220 | ¥448,220 | Social insurance fee |
| Pericardiotomy | | ¥595,641 | ¥595,641 | [32, 33] |
| BSC | | ¥632,100 | ¥632,100 | [34] |
| Discounting | | | | |
| Discount rate | | 3% | 3% | |

Abbreviations: PBT, proton beam therapy; 3D-CRT, three-dimensional conformal radiotherapy; BSC, best supportive care; RP, radiation pneumonitis; PE, pleural effusion; PCE, pericardial effusion

Grade 3, this was 0.026 in 3D-CRT and 0.0025 in PBT; for PE Grade 3, this was 0.06 in 3D-CRT and 0 in PBT; and for PCE Grade 3, this was 0.175 in 3D-CRT and 0.101 in PBT (combined with the overlapping health state).

Reflecting no difference in therapeutic response to adverse events between 3D-CRT and PBT, transition probabilities among RP grades were determined as described by Watanabe et al. [23], the probability between PE grades was assumed by expert opinion, and probabilities from PCE grades, including overlapping states, to subsequent, such as drainage, were set according to Pao et al. and Virk et al. and by expert opinion [24, 25].

Reflecting the assumption of no difference in OS and PFS, the probability of transitioning from any health state to BSC was calculated using OS data from the JCOG 9906 study on 3D-CRT [3], which was also applied to PBT. The transition from BSC to death was assumed to occur in one cycle based on expert opinion.

## 2.5 Utility

In this study, the quality-adjusted life year (QALY) was used as the evaluation index, and utility values were determined based on a literature review (Table 1). The QOL after radiotherapy for LAEC was based on a report by Sawada et al. [26]. For adverse events that were not obtained from Sawada et al, a literature search was conducted to obtain utility scores for similar diseases. For example, for RP Grade 3, we referred to the utility score during hospital-based treatment of pneumonia [27]; for PE Grade 2, we referred to the utility score of PE in lung cancer [28]; and for Grade 3, we referred to the utility score for the insertion of a thoracic drain [29]. For PCE Grade 3 and Grade 3 PCE + Grade 2 PE, we used a report of heart failure cases [30]. For BSC, we used the report of Nakamura et al. [31].

## 2.6 Cost calculation

Under Japan's social insurance system, the primary payers of medical costs are the social insurer and patient; therefore, medical costs paid directly by the social insurer and patient were considered.

Although not covered by social insurance, the cost of PBT was determined by referring to the fee for diseases already covered by social insurance in two base cases: 1,600,000 yen, referred to as the non-rare cancer base-case, and 2,375,000 yen, referred to as the rare cancer base-case. In addition, 3D-CRT was assumed to cost 727,600 yen (US$6,313) based on the current social insurance system. The fee in Japan's hospitalization system is based on the Diagnosis Procedure Combination (DPC), a comprehensive classification and evaluation system, and is calculated based on the number of hospital stays. However, the DPC includes the cost of anticancer drugs and imaging tests but not radiotherapy. Outpatient treatment costs were paid on a fee-for-service basis. These costs were calculated based on the opinions of radiotherapy specialists and the University of Tsukuba Hospital Medical Fee Claims Office.

Hospitalization costs for CRT were calculated assuming that a six-week hospitalization was required for complete chemoradiation and two subsequent hospitalizations for one week chemotherapy. The costs for follow-up and onset of each adverse event were calculated by listing the necessary medical procedures based on expert opinion and accumulating the medical procedures. Follow-up in asymptomatic patients included medical examination, blood sampling, chest radiography, contrast-enhanced computed tomography, and upper gastrointestinal endoscopy. Regarding the costs of adverse events, it was necessary to determine the number of days of hospitalization because the calculation was based on the DPC when treatment in hospital was necessary. Based on expert opinions, we estimated 21 and 11 days for RP Grade 3 and PE Grade 3, respectively, and 16 days for pericardial drainage. For pericardiotomy, the cost of preoperative examination, 11 days of hospitalization, and anesthesia were assumed based on literature reports [32, 33]; for BSC, the cost was calculated based on a report compiled by Ashino, assuming end-of-life home care [34].

## 2.7 Sensitivity analysis

To evaluate the stability of the ICER due to the assumptions of the economic model and explore the relative effect of each variable, we conducted a one-way sensitivity analysis and a probabilistic sensitivity analysis (PSA), that is, a Monte Carlo simulation of 1000 times. The lower and upper limits were set to vary by ±20% for cost, utility, and transition probability, and PSA distributions were triangular. The utility scores were adjusted so that, for example, Grade 1 utility was not less than Grade 2 utility.

As mentioned earlier, two base case analyses were conducted to determine the differences in fees for PBT. In the national fee schedule, two fee levels are set for rare and non-rare. These two base cases were used as equivalent base cases and no one-way sensitivity analysis was conducted. The treatment outcomes and costs were discounted at an annual rate of 3%. Model parameterization was performed using the TreeAge Pro software (version 2020; TreeAge, Inc., Williamstown, MA, USA).

## 3. Results

The results of the CUA for PBT compared with those for 3D-CRT are shown in Table 2. The standard treatment, 3D-CRT, resulted in 2.51 QALYs, whereas PBT resulted in 2.62 QALYs; the ICER per QALY was 14,025,268 yen (US$121,958) in the rare cancer base-case and 7,026,402 yen (US$61,099) in the non-rare cancer base-case.

Fig 2 shows the results of the one-way sensitivity analysis for both base cases, showing the top ten variables with sensitive ICERs from the base case. Nine of the ten items were utility scores, and only one was a transition probability (probability of Grade 3 PCE occurrence in 3D-CRT). The item with the most sensitive cost variation is also included in the figure as a reference (cost of pericardial drainage). The top ten variables between the two base cases did not

**Table 2. Cost-effectiveness of PBT and 3DCRT.**

(A) Rare cancer base-case

|  | Cost (\） | Incremental cost (\） | Effectiveness (QALY) | Incremental effectiveness (QALY) | ICER (\/QALY) |
|---|---|---|---|---|---|
| Proton beam therapy (PBT) | 4,468,532 | 1,553,049 | 2.6185 | 0.1107 | 14,025,268 |
| Conventional photon radiotherapy (3D-CRT) | 2,915,483 |  | 2.5078 |  |  |

(B) Non-rare cancer base-case

|  | Cost (\） | Incremental cost (\） | Effectiveness (QALY) | Incremental effectiveness (QALY) | ICER (\/QALY) |
|---|---|---|---|---|---|
| Proton beam therapy (PBT) | 3,693,532 | 778,049 | 2.6185 | 0.1107 | 7,026,402 |
| Conventional photon radiotherapy (3D-CRT) | 2,915,483 |  | 2.5078 |  |  |

Abbreviations: QALY, quality-adjusted life year; ICER, incremental cost-effectiveness ratio; 3D-CRT, three-dimensional conformal radiotherapy

differ, and the most sensitive item was the utility score for no late adverse events. The impact of the most sensitive item on transition probability and cost was negligible.

Fig 3 shows the cost-effectiveness acceptability curve. Using 1000 ICERs generated by Monte Carlo simulations, the probability of the ICER falling below 7, 500, 000 yen (US $65,217) was 20.1% for the rare cancer base-case and 55.1% for the non-rare cancer base-case.

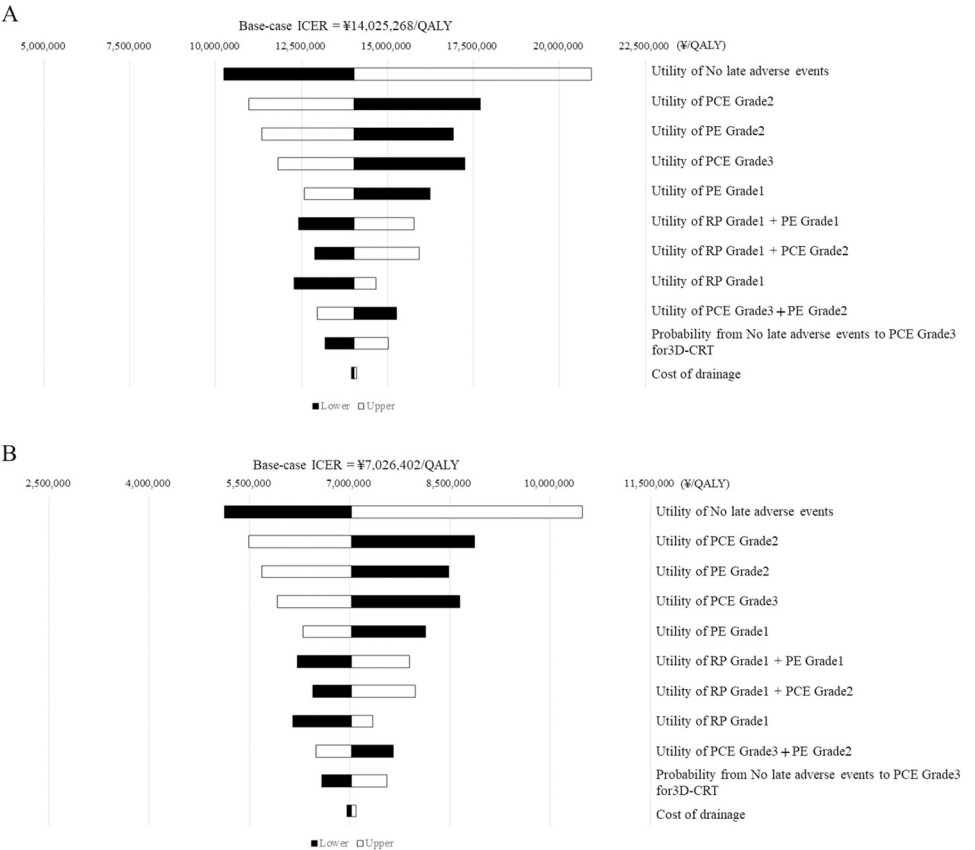

**Fig 2. Results of one-way sensitivity analysis.** (A) Rare cancer base-case. (B) Non-rare cancer base-case. Abbreviations: PCE, pericardial effusion; PE, prepupal effusion; 3D-CRT, three-dimensional conformal radiotherapy. Ten variables with large incremental cost-effectiveness ratio (ICER) changes from the base case values are shown. The ICERs were all >\7,500,000/QALY in the rare cancer base-case, and eight items in the non-rare cancer base-case were > \7,500,000/QALY.

A

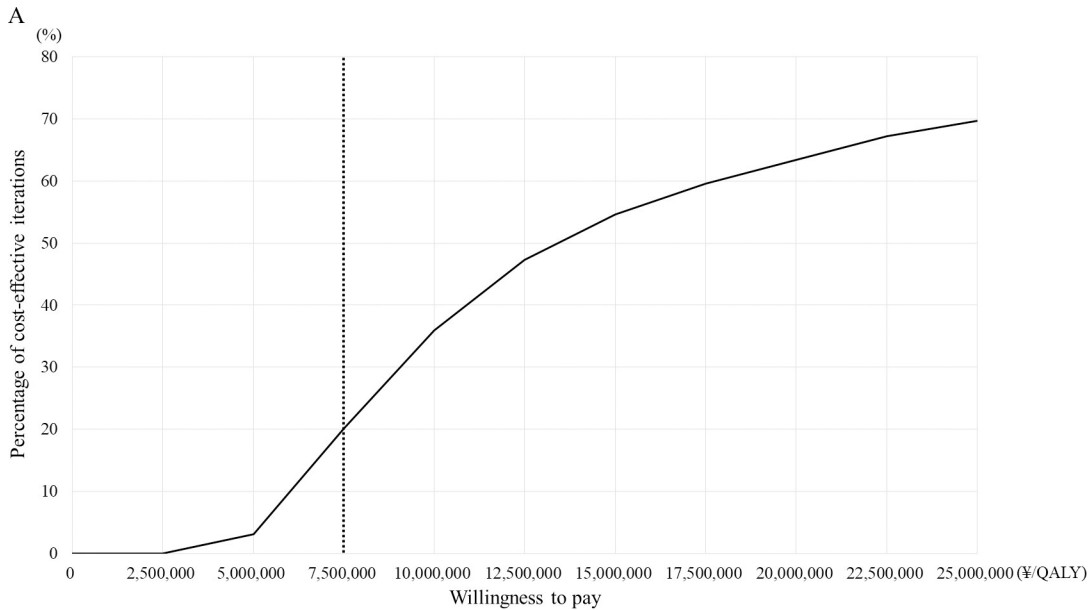

B

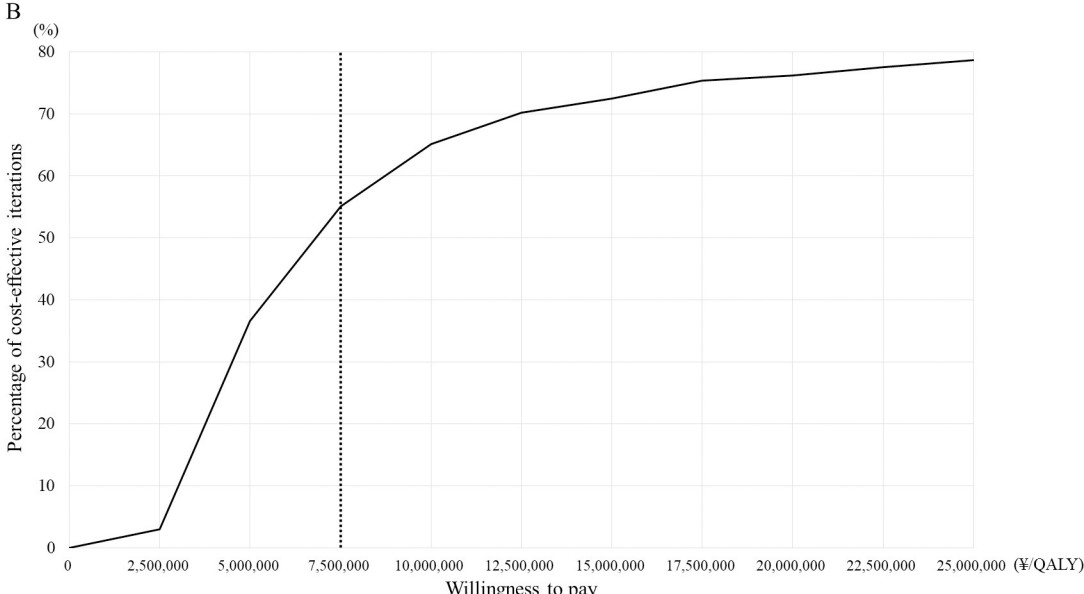

**Fig 3. Results of probabilistic sensitivity analyses: cost-effectiveness acceptability curve.** (A) Rare cancer base-case. (B) Non-rare cancer base-case. Among the 1,000 incremental cost-effectiveness ratios (ICERs) produced by the Monte Carlo simulation, the probability that the ICER was less than ¥7,500,000 (US$65,217) per quality-adjusted life year (QALY) gained was 20.1% in the rare cancer base-case and 55.1% in the non-rare cancer base-case.

## 4. Discussion

The cost-effectiveness of PBT as an alternative to 3D-CRT for LAEC was examined and the ICER was estimated to be 14,025,268 yen/QALY for rare cancer base-case and 7,026,402 yen/QALY for non-rare cancer base-case. In light of the threshold set by Japan's Central Social Insurance Medical Council for the social willingness to pay for innovative cancer care, which is 7,500,000 yen for one QALY gain, the former is judged as not cost-effective, whereas the latter is cost-effective. All ICERs produced by the one-way sensitivity analysis for the rare cancer

base-case were greater than the threshold. Conversely, for the non-rare cancer base-case, changes in nine variables could increase ICERs to a level greater than the threshold while probabilistic sensitivity analysis showed that the probability of being less than the threshold was 55.1%. Therefore, the cost-effectiveness of PBT for LAEC depends on the difference between the two base cases, i.e., the fee level under social insurance. At a lower fee level, it could be cost-effective, and its inclusion in the benefits package is justifiable by the straightforward application of the decision rule.

The proposed model performed reasonably well. The directions of the ICER changes corresponding to the assumed changes in the one-way sensitivity analysis were anticipated. The higher sensitivities found among the utility scores are consistent with our modeling approach in that the advantages of PBT are incorporated by focusing on adverse event differences.

Regarding the fee settings for PBT under social insurance, rare cancers with higher fees include liver cancer with a diameter of 4 cm or larger, locally advanced unresectable pancreatic cancer, intrahepatic bile duct cancer, and locally recurrent colorectal cancer in the gastrointestinal field. Conversely, prostate cancer is the only non-rare cancer with a lower fee. Although there is no explicit criterion that distinguishes these cancers, the incidence rate of prostate cancer is approximately four times higher than that of liver and pancreatic cancers. The incidence rate of esophageal cancer is similar to that of liver and pancreatic cancers in women [35]. Fixing a higher fee for LAEC is consistent with past practices. The technical expertise required for PBT for LAEC is much higher than that for prostate cancer; PBT for LAEC deserves a higher fee, which would make PBT unavailable to patients according to cost-effectiveness judgments. To benefit patients, applying a lower fee should be considered.

To the best of our knowledge, a cost-effective analysis of PBT for esophageal cancer has not yet been reported. However, the results of this analysis should be carefully considered for use in other healthcare systems because the expected costs vary between countries. In addition, benefits may also vary from country to country. For example, radiation doses differ between East Asian countries, such as Japan and China, and Western countries. In East Asia, the standard radiation dose is 60 gray (Gy) because squamous cell carcinoma is common [36, 37], whereas in Western countries, adenocarcinoma is the main type of cancer and 50.4 Gy is the standard [2]. In general, the higher the irradiation dose, the stronger the adverse events. Therefore, a dose of 50.4 Gy may be unfavorable for cost-effectiveness.

The prevalence of IMRT in the treatment of esophageal cancer also differs from country to country. In Japan, the latest radiotherapy guidelines mention IMRT, but 3D-CRT is listed as the standard treatment modality [4]. The percentage of esophageal cancer patients treated with IMRT in Japan has been on the rise since 2016, when statistical data were available, but has remained in the 10% range even in the most recent years [38]. Depending on the future popularity of IMRT, cost-effective analysis comparing IMRT and PBT may become more valuable, but the percentage of pancreatic cancer patients using IMRT, for example, has been reported to have leveled off since 2018, and future trends need to be closely monitored to see if IMRT will become the standard treatment [39].

Future studies may resolve the debate on whether the OS for PBT is superior to that for 3D-CRT. In this study, we assumed that OS and PFS does not differ between 3D-CRT and PBT and did not consider underlying diseases, such as cardiac or pulmonary disease, in the patient background. The presence of an underlying disease may further strengthen the potential cost-effectiveness of PBT in reducing adverse events, and if PBT is shown to improve OS, the ICER could become favorable. In addition, a search of PubMed for cost-effectiveness analyses for PBT in Japan found three reports. Two were in pediatric medulloblastoma, focusing on hearing impairment and second cancer risk, respectively [40, 41]. The other was in pancreatic cancer [39]. All reports indicated that PBT was cost-effective.

This study had several limitations. First, the clinical evidence supporting the reduction in adverse events with PBT was based on retrospective studies. However, a prospective study of PBT and carbon-ion radiotherapy in Japan also reported the frequency of RP and PCE Grade 3 or higher, which is similar to the frequency of occurrence used in this study [42]. Second, we did not model secondary or subsequent treatments related to recurrence because the analysis was based on the assumption that PFS, OS and accompanying costs do not differ. Third, for several utility scores, we used reports from other diseases with similar patient conditions, and from other countries. This is because not all utility scores were available in the literature search. Finally, this study focused on the cost-effectiveness of PBT as an alternative to 3D-CRT in existing facilities and did not calculate the cost of building new facilities because PBT is included in social insurance coverage for several cancers in Japan. Capital costs should be considered in healthcare systems that require the construction of new facilities, and the share and overhead costs of PBT for other cancers should also be considered.

## 5. Conclusions

From a health economics standpoint, PBT is a cost-effective alternative to 3D-CRT for LAEC if it is categorized as a non-rare cancer case in the social insurance fee schedule, and it is justifiable to make it available to patients as part of the social insurance benefit package in Japan.

## Acknowledgments

The authors are grateful to Shinji Sugahara, MD, PhD. Department of Radiology, Tokyo Medical University Ibaraki Medical Center, for providing expert opinions on the study. The authors would also like to thank the Social Insurance Claims Group, Division of Strategic Management of the University of Tsukuba Hospital for their advice regarding the medical expenses examined in this study.

## Author Contributions

**Conceptualization:** Takuya Sawada, Masahide Kondo, Yuichi Hiroshima.

**Data curation:** Takuya Sawada, Masahide Kondo, Masaaki Goto, Yuichi Hiroshima.

**Formal analysis:** Takuya Sawada, Masahide Kondo, Yuichi Hiroshima, Shu-Ling Hoshi.

**Funding acquisition:** Hideyuki Sakurai.

**Investigation:** Takuya Sawada, Masahide Kondo, Shu-Ling Hoshi.

**Methodology:** Takuya Sawada, Masahide Kondo, Shu-Ling Hoshi.

**Project administration:** Toshiyuki Okumura, Hideyuki Sakurai.

**Supervision:** Toshiyuki Okumura, Hideyuki Sakurai.

**Visualization:** Takuya Sawada, Masahide Kondo.

**Writing – original draft:** Takuya Sawada, Masahide Kondo, Masaaki Goto, Motohiro Murakami, Toshiki Ishida, Yuichi Hiroshima, Shu-Ling Hoshi, Reiko Okubo, Toshiyuki Okumura, Hideyuki Sakurai.

**Writing – review & editing:** Takuya Sawada, Masahide Kondo, Masaaki Goto, Motohiro Murakami, Toshiki Ishida, Yuichi Hiroshima, Shu-Ling Hoshi, Reiko Okubo, Toshiyuki Okumura, Hideyuki Sakurai.

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
