## [Decision Letter · Decision Letter 0]

17 Apr 2024

PONE-D-24-04902Cost-Utility Analysis of Proton Beam Therapy for Locally Advanced Esophageal Cancer in JapanPLOS ONE

Dear Dr. Masahide,

Thank you for submitting your manuscript to PLOS ONE. After careful consideration, we feel that it has merit but does not fully meet PLOS ONE’s publication criteria as it currently stands. Therefore, we invite you to submit a revised version of the manuscript that addresses the points raised during the review process. **Please carefully follow the queries raised by esteemed reviewers which also are major, to revise the manuscript and provide appropriate responses or justifications.**

We look forward to receiving your revised manuscript.

Kind regards,

Sina Azadnajafabad, MD, MPH

Academic Editor

PLOS ONE

Journal Requirements:

Reviewers' comments:

Reviewer's Responses to Questions

**Comments to the Author**

1. Is the manuscript technically sound, and do the data support the conclusions?

Reviewer #1: Partly

Reviewer #2: Yes

Reviewer #3: No

2. Has the statistical analysis been performed appropriately and rigorously? 

Reviewer #1: Yes

Reviewer #2: Yes

Reviewer #3: Yes

3. Have the authors made all data underlying the findings in their manuscript fully available?

Reviewer #1: Yes

Reviewer #2: Yes

Reviewer #3: Yes

4. Is the manuscript presented in an intelligible fashion and written in standard English?

Reviewer #1: Yes

Reviewer #2: Yes

Reviewer #3: Yes

5. Review Comments to the Author

**Reviewer #1**: #1 “In Japan, 3D CRT is the standard radiotherapy method with strong evidence for definitive CRT for LAEC [3, 4].” & “In addition, the Japanese guidelines for the treatment of esophageal cancer recommend PBT over IMRT as the treatment of choice for patients with poor cardiopulmonary function [9]” while ref-3 & ref-4 & ref-9 were papers from Japan.” & “We compared PBT with the standard treatment, 3D-CRT”

>> Please note that another guideline from Japan published in 2022 (ref-1 of the current manuscript) stated “Recently, high-precision radiotherapy, such as intensity-modulated radiotherapy with X-rays and particle therapy with proton or heavy ion beams, has been reported to reduce these adverse effects” without clear statement that that PBT was superior to IMRT。A small randomized controlled trial had reported significantly lower normal organ dose when IMRT was compared to 3DCRT [https://pubmed.ncbi.nlm.nih.gov/21764711/ ]. Furthermore, IMRT was the sole treatment modality in a recent landmark study from China [Int J Radiat Oncol Biol Phys. 2023 Apr 1;115(5):1129-1137] and was advocated in another recent landmark study from Dutch [J Clin Oncol . 2021 Sep 1;39(25):2816-2824]. IMRT was also preferred [over 3DCRT] by the European guideline [ref-2 of the current manuscript] according to its statement “RT should be delivered using 3D conformal RT, but intensity modulated RT or volumetric arc therapy are preferred to better minimise the radiation dose to critical normal tissues.”. The American guideline [NCCN Esophageal cancer 2024v1] also stated “IMRT is now standardly used in the preoperative, definitive, and postoperative treatment of esophageal and esophagogastric cancer”.

The reviewer is not a Japanese and is not sure if 3DCRT (rather than IMRT) was really the standard of radiotherapy modality in Japan, but even so, the results of this study [comparing proton vs 3DCRT] will not be generalizable to many other countries where IMRT was the mainstream of radiotherapy modality for esophageal cancer.

#2 “Adverse Events ([CTCAE] ver. 4.0) adverse events of RP, PE, and PCE were reported to be 9%, 16%, and 135 4%, respectively, for 3D-CRT [3] and 0.5%, 0%, and 1%, respectively, for PBT [5].” >> Adverse Events (CTCAE ver. 4.0) of RP, PE, and PCE were reported to be 9%, 16%, and 135 4%, respectively, for 3D-CRT [3] and 0.5%, 0%, and 1%, respectively, for PBT [5].? Furthermore, please justify and clarify. Please note CTCv2 was used in ref-3 & CTC v5 was in use currently [https://ctep.cancer.gov/protocoldevelopment/electronic_applications/ctc.htm#ctc_60 accessed 2024/3/22]

#3 “Every patient with or without adverse events was moved to best supportive care (BSC), followed by death” >> This was not compatible with the Japanese guideline published in 2023 [Esophagus. 2023 Jul;20(3):343-372] Fig 2 in which salvage therapy was used for residual/recurrence.

#4 “the clinical evidence supporting the reduction in adverse events with PBT was based on retrospective studies” >> suggest to comment on randomized trials such as J Clin Oncol. 2020 May 10;38(14):1569-1579, although vs IMRT and out of Japan

#5 Fig 1. What is the content of fig 1b Markov model? Not seen in the manuscript.

#6 Suggest to provide some kind of checklist if not yet [such as CHEERS 2022, Pharmacoeconomics. 2022 Jun;40(6):601-609] as an supplement to ensure the readers regarding the quality of the current manuscript

**Reviewer #2**: In this paper, the authors conduct a cost effectiveness study of PBT in Japan, finding that PBT is cost effective for advanced esophagus cancer in the lower cost schema, but not in the higher cost schema. The study is really cleanly done, very well written, and tempered in its conclusions. The authors should be commended for their work. I have a few minor suggestions:

1. Perhaps highlight in the limitations that the OS benefit for PBT was really only based on one study, that is, Ono et al, in Ref 5. This is fine, but should be noted as a limitation, since their findings and the utility of PBT from OS standpoint are hinged on this one study.

2. The authors make an important observation in the discussion that cost-effectiveness may be different if 5040cGy were used commonly rather than 6000cGy, given the differences in epidemiology in Japan. This is an insightful point that the authors bring up. I think the authors can take this a step further and add to the discussion some literature review on 1) cost-effectiveness of PBT for esophagus cancer in in neighboring countries, and 2) cost-effectiveness of PBT for other diseases in Japan and in other nearby countries.

Great work!!

**Reviewer #3:** This paper evaluated the cost-effectiveness of Proton beam therapy (PBT) as a replacement for conventional three-dimensional conformal radiotherapy (3D-CRT) for locally advanced esophageal cancer (LAEC) that is not covered by social insurance, and authors demonstrated that the inclusion of PBT for LAEC in the benefit package of social insurance is cost-effective if a lower fee is applied. Finally, authors concluded that PBT is a cost-effective alternative to 3D-CRT for LAEC and making it available to patients under social insurance could be justifiable. This study dealt with important issue regarding the cost-effectiveness of PBT. However, the design of this study seemed to evaluate the threshold at which PBT is more cost-effective than 3D-CRT for LAEC. The cost calculations should include the expenses for PBT equipment, its reimbursement, and maintenance costs, which are not considered in this study. Therefore, I believe that the conclusions stated by the authors cannot be derived from this study design.

1) Authors assumed two base cases depending on the two existing levels of fees for PBT in social insurance: 2,735,000 Japanese yen (US$20,652) or 1,600,000 yen (US$13,913). However, PBT of LAEC has been not yet covered by national insurance and is still undergoing under the Advanced Medical Care. This cost setting does not reflect the actual costs of PBT. Why did authors assume these cost setting in evaluating the cost-effectiveness?

2) It is easily speculate that running cost of PBT would be higher than 3D-CRT, indicating that this will affect the cost-effectiveness between PBT and 3D-CRT.

6. PLOS authors have the option to publish the peer review history of their article (what does this mean?). If published, this will include your full peer review and any attached files.

Reviewer #1: No

Reviewer #2: No

Reviewer #3: No

---

## [Author Response · Author response to Decision Letter 0]

26 May 2024

We have submitted our replies to the editors and reviewers in a new attachment.

---

## [Decision Letter · Decision Letter 1]

14 Jun 2024

PONE-D-24-04902R1Cost-Utility Analysis of Proton Beam Therapy for Locally Advanced Esophageal Cancer in JapanPLOS ONE

Dear Dr. Masahide,

Thank you for submitting your manuscript to PLOS ONE. After careful consideration, we feel that it has merit but does not fully meet PLOS ONE’s publication criteria as it currently stands. Therefore, we invite you to submit a revised version of the manuscript that addresses the points raised during the review process.

We look forward to receiving your revised manuscript.

Kind regards,

Sina Azadnajafabad, MD, MPH

Academic Editor

PLOS ONE

Journal Requirements:

Reviewers' comments:

Reviewer's Responses to Questions

**Comments to the Author**

1. If the authors have adequately addressed your comments raised in a previous round of review and you feel that this manuscript is now acceptable for publication, you may indicate that here to bypass the “Comments to the Author” section, enter your conflict of interest statement in the “Confidential to Editor” section, and submit your "Accept" recommendation.

Reviewer #1: (No Response)

Reviewer #2: All comments have been addressed

2. Is the manuscript technically sound, and do the data support the conclusions?

Reviewer #1: Yes

Reviewer #2: Yes

3. Has the statistical analysis been performed appropriately and rigorously? 

Reviewer #1: Yes

Reviewer #2: Yes

4. Have the authors made all data underlying the findings in their manuscript fully available?

Reviewer #1: Yes

Reviewer #2: Yes

5. Is the manuscript presented in an intelligible fashion and written in standard English?

Reviewer #1: Yes

Reviewer #2: Yes

6. Review Comments to the Author

**Reviewer #1: **#1. Generalizability issue [CHEERS 2022 item 26]

Regarding my previous comment-1, the authors had provided references [such as J Radiat Res. 2022 Jul; 63(4): 646-656] to demonstrate that 3DCRT (rather than IMRT) was still the standard radiotherapy modality in Japan.

The reviewer agreed with the authors that there was a lack of high level evidence comparing IMRT vs 3DCRT, although a small randomized trial had reported (>= grade III radiation pneumonitis) was only seen in 3DCRT arm but not IMRT arm [https://pubmed.ncbi.nlm.nih.gov/21764711/]. Therefore, the reviewer agreed that (IMRT vs 3DCRT) was still a subject of research as mentioned by the reference [such as the above J Radiat Res. 2022 Jul; 63(4): 646-656] provided by the authors.

However, the reviewer would also like to point out the IMRT was already the main treatment modality in many countries other than Japan [references see my previous comment-1]. Therefore, the results of the current manuscript [comparing proton vs 3DCRT] will not be generalizable to these countries.

#2 The reviewer assumed fig 1-3 were the same as in version-1 because fig 1-3 were not seen in version-2.

**Reviewer #2:** Thank you for addressing my suggestions. The paper is in excellent shape and is an important contribution to the literature.

7. PLOS authors have the option to publish the peer review history of their article (what does this mean?). If published, this will include your full peer review and any attached files.

Reviewer #1: No

Reviewer #2: No

---

## [Author Response · Author response to Decision Letter 1]

19 Jul 2024

The comments from the Editor and reviewer are attached in the attached Word document. Please take a moment to review them.

---

## [Decision Letter · Decision Letter 2]

5 Aug 2024

Cost-Utility Analysis of Proton Beam Therapy for Locally Advanced Esophageal Cancer in Japan

PONE-D-24-04902R2

Dear Dr. Masahide,

We’re pleased to inform you that your manuscript has been judged scientifically suitable for publication and will be formally accepted for publication once it meets all outstanding technical requirements.

During the review process one of the reviewers suggested rejection of this submission for publication; however, by going through the responses of authors to the reviewer's comments and by seeking additional inspectors, we came with the accept decision on this submission.

Kind regards,

Sina Azadnajafabad, MD, MPH

Academic Editor

PLOS ONE

Additional Editor Comments (optional):

Reviewers' comments:

Reviewer's Responses to Questions

**Comments to the Author**

1. If the authors have adequately addressed your comments raised in a previous round of review and you feel that this manuscript is now acceptable for publication, you may indicate that here to bypass the “Comments to the Author” section, enter your conflict of interest statement in the “Confidential to Editor” section, and submit your "Accept" recommendation.

Reviewer #1: All comments have been addressed

2. Is the manuscript technically sound, and do the data support the conclusions?

Reviewer #1: Yes

3. Has the statistical analysis been performed appropriately and rigorously? 

Reviewer #1: Yes

4. Have the authors made all data underlying the findings in their manuscript fully available?

Reviewer #1: Yes

5. Is the manuscript presented in an intelligible fashion and written in standard English?

Reviewer #1: Yes

6. Review Comments to the Author

Reviewer #1: The authors responded to my questions. The reviewer had no additional comments. Best wishes for this manuscript.

7. PLOS authors have the option to publish the peer review history of their article (what does this mean?). If published, this will include your full peer review and any attached files.

Reviewer #1: No

---

## [Editor Report · Acceptance letter]

18 Sep 2024

PONE-D-24-04902R2 

PLOS ONE

Dear Dr. Kondo, 

I'm pleased to inform you that your manuscript has been deemed suitable for publication in PLOS ONE. Congratulations! Your manuscript is now being handed over to our production team.

Kind regards, 

on behalf of

Dr. Sina Azadnajafabad 

Academic Editor

PLOS ONE